# Mol-CycleGAN - a generative model for molecular optimization

## Abstract

Designing a molecule with desired properties is one of the biggest challenges in drug development, as it requires optimization of chemical compound structures with respect to many complex properties. To augment the compound design process we introduce Mol-CycleGAN – a CycleGAN-based model that generates optimized compounds with a chemical scaffold of interest. Namely, given a molecule our model generates a structurally similar one with an optimized value of the considered property. We evaluate the performance of the model on selected optimization objectives related to structural properties (presence of halogen groups, number of aromatic rings) and to a physicochemical property (penalized logP). In the task of optimization of penalized logP of drug-like molecules our model significantly outperforms previous results.

## 1 Introduction

The principal goal of the drug design process is to find new chemical compounds that are able to modulate the activity of a given target (typically a protein) in a desired way (Ratti & Trist, 2001). However, finding such molecules in high-dimensional chemical space of all molecules without any prior knowledge is nearly impossible. *In silico* methods have been introduced to leverage the existing chemical, pharmacological and biological knowledge, thus forming a new branch of science - computer-aided drug design (CADD) (Rao & Srinivas, 2011; Bajorath, 2002). In particular, the recent advancements in deep learning encouraged its application to CADD (Chen et al., 2018). Computer methods are nowadays applied at every stage of drug design pipelines (Rao & Srinivas, 2011) - from the search of new, potentially active compounds (Lavecchia & Di Giovanni, 2013), through optimization of their activity and physicochemical profile (Honório et al., 2013) and simulating their scheme of interaction with the target protein (de Ruyck et al., 2016), to assisting in planning the synthesis and evaluation of its difficulty (Segler et al., 2018).

In the center of our interest are the hit-to-lead and lead optimization phases of the compound design process. Their goals are to optimize drug-like molecules identified in previous steps in terms of, respectively, the desired activity profile (increased potency towards given target protein and provision of inactivity towards undesired proteins) and the physicochemical and pharmacokinetic properties. The challenge here is to optimize a molecule with respect to multiple properties simultaneously (Honório et al., 2013).

Our principal contribution is the introduction of Mol-CycleGAN, a generative model based on CycleGAN (Zhu et al., 2017) with the goal to augment the compound design process. We show that our model can generate molecules that possess desired properties[1] while retaining their chemical scaffolds. Given a starting molecule, the model generates a similar one but with a desired characteristics. The similarity between the two molecules is important in the context of multiparameter optimization, as it makes it easier to optimize the selected property without spoiling the previously optimized ones. To the best of our knowledge, this is the first approach to molecule generation that uses the CycleGAN architecture.

We evaluate our model on its ability to perform structural transformations and molecular optimization. The former indicates that the model is able to do simple structural modifications such as a change in the presence of halogen groups or number of aromatic rings. In the latter, we aim to

---

[1]By molecular property we also mean binding affinity towards target protein.

maximize penalized logP to assess the model's utility for compound design. Penalized logP is a physicochemical property often selected as a testing ground for molecule optimization models (Jin et al., 2018; You et al., 2018), as it is relevant in the drug design process. In the optimization of penalized logP for drug-like molecules our model significantly outperforms previous results.

## 2 RELATED WORK

There are two main approaches of applying deep learning in drug design (see Chen et al. (2018) for a recent review): ($a$) use discriminative models to screen commercial databases and classify molecules as likely active or likely inactive (virtual screening); ($b$) use generative models to propose novel molecules that likely possess the desired properties. The former application already proved to give outstanding results (Duvenaud et al., 2015; Jastrzębski et al., 2016; Coley et al., 2017; Pham et al., 2018). The latter use case is rapidly emerging.

Many generative deep learning models have already been applied in the compound design context. Initial molecule generation models mostly operate on SMILES strings (Weininger, 1988). Long short-term memory (LSTM) network architecture is applied in Segler et al. (2017); Bjerrum & Threlfall (2017); Winter et al. (2018); Gupta et al. (2018). **Variational Autoencoder (VAE)** (Kingma & Welling, 2013) is used by Gómez-Bombarelli et al. (2018) to generate SMILES of new molecules. Unfortunately, these models can generate invalid SMILES that do not correspond to any molecules. Introduction of grammars into the model improved the success rate of valid SMILES generation (Kusner et al., 2017; Dai et al., 2018). Maintaining chemical validity within a generative process became possible through VAEs realized directly on molecular graphs (Simonovsky & Komodakis, 2018; Jin et al., 2018).

**Generative Adversarial Networks (GANs)** (Goodfellow et al., 2014) are an alternative architecture that has been applied to *de novo* drug design. Guimaraes et al. (2017) propose GANs and Reinforcement Learning (RL) model (based on SMILES), which generates samples that fulfill desired objectives while promoting diversity. De Cao & Kipf (2018) use GANs and RL, together with graph representation (adjacency and annotation matrices) to generate new molecules with the given properties. You et al. (2018) train convolutional GANs on molecular graphs and use RL to ensure that the proposed molecules have logP and molecular weight in the desired range.

Indeed, some of these models can be used to effectively search through the chemical space. Nevertheless, these approaches are not without flaws. The generated compounds can be, e.g., difficult or impossible to synthesize. We address this issue by proposing Mol-CycleGAN, a generative model designed to generate molecules with the desired properties while retaining their chemical scaffolds. Such a model can prove to be very useful for optimizing active molecules towards a given property, which is essential in compound design.

## 3 MOL-CYCLEGAN

We introduce Mol-CycleGAN to perform optimization by learning from the *sets* of molecules with or without the desired molecular property (denoted by the sets $X$ and $Y$). Our approach is to train a model to perform the transformation $G : X \rightarrow Y$ and then use this model to perform optimization of molecules. In the context of compound design $X$ ($Y$) can be, e.g., the set of inactive (active) molecules.

To represent the sets $X$ and $Y$ our approach requires an embedding of molecules, from which it should be possible to decode the coordinates back into some complete representation (e.g., the SMILES representation). Here, the latent space of variational autoencoders can be used. This has the added benefit that the distance between molecules (required to calculate the loss function) can be defined in the latent space. Essential chemical properties are easier to express on graphs rather than linear SMILES representations (Weininger, 1988). This is why for molecule representation we use the latent space obtained from Junction Tree Variational Autoencoder (JT-VAE) (Jin et al., 2018). JT-VAE is based on a graph structure of molecules and shows superior properties compared to SMILES-based VAEs (cf. also the discussion in Section 2). One could also try formulating the CycleGAN on the SMILES representation directly, but this would raise the problem of defining

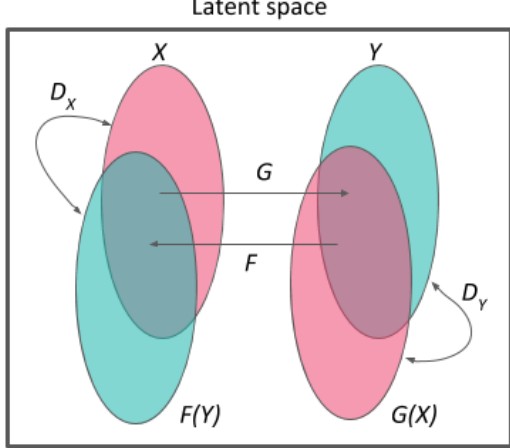

Figure 1: Schematic diagram of our Mol-CycleGAN. $X$ and $Y$ are the sets of molecules with selected values of the molecular property (e.g. active/inactive or with high/low values of logP). $G$ and $F$ are the generators. $D_X$ and $D_Y$ are the discriminators.

the intermolecular distance, as the standard manners of measuring similarity between molecules (Tanimoto similarity) are non-differentiable.

Our approach extends the CycleGAN framework (Zhu et al., 2017) to molecular embeddings, created by JT-VAE (Jin et al., 2018). We represent each molecule with a latent vector, given by the mean of the variational encoding distribution. The inclusion of the cyclic component acts as a regularization and may also help in the regime of low data, as the model can learn from both directions of the transformation. With the cyclic component the resulting model is more robust (cf. e.g. the comparison of non-cyclic IcGAN (Perarnau et al., 2016) vs CycleGAN in Choi et al. (2017)). Our model works as follows (cf. Fig. 1): (i) we start by defining the sets $X$ and $Y$ (e.g., active/inactive molecules); (ii) we introduce the mapping functions $G : X \to Y$ and $F : Y \to X$; (iii) we introduce discriminators $D_X$ (and $D_Y$) which force the generator $F$ (and $G$) to generate samples from a distribution close to the distribution of $X$ (or $Y$). The components $F$, $G$, $D_X$, and $D_Y$ are implemented with neural networks acting in the latent space (see Appendix A for technical details).

After training the model we perform optimization of a given molecule by: (i) computing its latent space embedding, $x$; (ii) using the generating function to compute $G(x)$; (iii) decoding the latent space coordinates given by $G(x)$ to obtain the SMILES representation of the optimized molecule. Thereby, any molecule can be cast onto the set of molecules with the desired property, $Y$.

For training the Mol-CycleGAN we use the following loss function:

$$L(G, F, D_X, D_Y) = L_{\text{GAN}}(G, D_Y, X, Y) + L_{\text{GAN}}(F, D_X, Y, X)$$
$$+ \lambda_1 L_{\text{cyc}}(G, F) + \lambda_2 L_{\text{identity}}(G, F),$$

and aim to solve

$$G^*, F^* = \arg \min_{G,F} \max_{D_X,D_Y} L(G, F, D_X, D_Y).$$

We use the adversarial loss introduced in LS-GAN (Mao et al., 2017)

$$L_{\text{GAN}}(G, D_Y, X, Y) = \frac{1}{2} \mathbb{E}_{y \sim p_{\text{data}}(y)}[(D_Y(y) - 1)^2] + \frac{1}{2} \mathbb{E}_{x \sim p_{\text{data}}(x)}[(D_Y(G(x)))^2],$$

which ensures that the generator $G$ (and $F$) generates samples from a distribution close to the distribution of $Y$ (or $X$). The cycle consistency loss

$$L_{\text{cyc}}(G, F) = \mathbb{E}_{y \sim p_{\text{data}}(y)}[\|G(F(y)) - y\|_1] + \mathbb{E}_{x \sim p_{\text{data}}(x)}[\|F(G(x)) - x\|_1]$$

is responsible for reducing the space of possible mapping functions, such that for a molecule $x$ from set $X$, the GAN cycle brings it back to a molecule similar to $x$, i.e. $F(G(x))$ is close to $x$ (and

analogously $G(F(y))$ is close to $y$. Finally, to ensure that the optimized molecule is close to the starting one we use the identity mapping loss (Zhu et al., 2017)

$$L_{\text{identity}}(G, F) = \mathbb{E}_{y \sim p_{\text{data}}(y)}[\|F(y) - y\|_1] + \mathbb{E}_{x \sim p_{\text{data}}(x)}[\|G(x) - x\|_1],$$

which further reduces the space of possible mapping functions and prevents the model from generating molecules that lay far away from the starting molecule in the JT-VAE latent space.

In all our experiments we use the values of hyperparameters $\lambda_1 = 0.3$ and $\lambda_2 = 0.1$, which were chosen by checking a couple of combinations (for structural tasks) and verifying that our optimization process: (i) improves the studied property and (ii) generates molecules similar to the starting ones. We have not performed a grid search for optimal values of $\lambda_1$ and $\lambda_2$, and hence there could be space for improvement here. Note that these parameters control the balance between improvement in the optimized property and similarity between the generated and the starting molecule. Both the improvement and the similarity can be obtained with our model, as we show in the next section.

## 4 RESULTS

We conduct experiments to test whether the proposed model is able to generate molecules that possess desired properties and are close to the starting molecules. Namely, we evaluate the model on tasks related to structural modifications, as well as on tasks related to molecule optimization. For testing molecule optimization we select the octanol-water partition coefficient (logP) penalized by the synthetic accessibility (SA) score. logP describes lipophilicity - a parameter influencing a whole set of other characteristics of compounds such as solubility, permeability through biological membranes, ADME (absorption, distribution, metabolism, and excretion) properties, and toxicity. We use the formulation as in Jin et al. (2018) (see Appendix D therein). Explicitly, for molecule $m$ the penalized logP is given as $logP(m) - SA(m)$. We use the ZINC-250K dataset used earlier by Kusner et al. (2017); Jin et al. (2018) which contains 250 000 drug-like molecules extracted from the ZINC database (Sterling & Irwin, 2015). Molecular similarity and drug-likeness are achieved in all experiments. The detailed formulation of the tasks is the following:

- **Structural transformations** We test the model's ability to perform simple structural transformations of the molecules:
    - **Halogen moieties** We split the dataset into two subsets $X$ and $Y$. The set $Y$ consists of molecules which contain at least one of the following SMARTS: '[!#1]Cl', '[!#1]F', '[!#1]I', 'C#N', whereas the set $X$ consists of such molecules which do not contain any of them. The SMARTS chosen in this experiment indicate halogen moieties and the nitrile group. Their presence and position within a molecule can have an immense impact on the compound's activity.
    - **Aromatic rings** Molecules in $X$ have exactly two aromatic rings, whereas molecules in $Y$ have one or three aromatic rings.
- **Constrained molecule optimization** We optimize penalized logP, while constraining the degree of deviation from the original molecule. The similarity between molecules is measured with Tanimoto similarity on Morgan Fingerprints (Rogers & Hahn, 2010). The set $X$ ($Y$) is a random sample from ZINC-250K of the compounds with penalized logP below (above) median. Here we follow the task previously proposed in Jin et al. (2018).
- **Unconstrained molecule optimization** We perform unconstrained optimization of penalized logP. The set $X$ is a random sample from ZINC-250K and the set $Y$ is a random sample from the top-20% molecules with the highest penalized logP in ZINC-250K.

### 4.1 STRUCTURAL TRANSFORMATIONS

In each structural experiment, we test the model's ability to perform simple transformations of molecules in both directions $X \to Y$ and $Y \to X$. Here, $X$ and $Y$ are non-overlapping sets of molecules with a specific structural property. We start with experiments on structural properties because they are easier to interpret and the rules related to transforming between $X$ and $Y$ are well defined. Hence, the present task should be easier for the model, as compared to the optimization of complex molecular properties, for which there are no simple rules connecting $X$ and $Y$.

Table 1: Evaluation of models modifying the presence of halogen moieties and the number of aromatic rings. Success rate is the fraction of times when a desired modification occurs. Non-identity is the fraction of times when the generated molecule is different from the starting one. Uniqueness is the fraction of unique molecules in the set of generated molecules.

| | Halogen moieties | | Aromatic rings | |
| --- | --- | --- | --- | --- |
| | $X \to G(X)$ | $Y \to F(Y)$ | $X \to G(X)$ | $Y \to F(Y)$ |
| Success rate | 0.6429 | 0.7161 | 0.5342 | 0.4216 |
| Non-identity | 0.9345 | 0.9574 | 0.9082 | 0.8899 |
| Uniqueness | 0.9952 | 0.9953 | 0.9957 | 0.9954 |

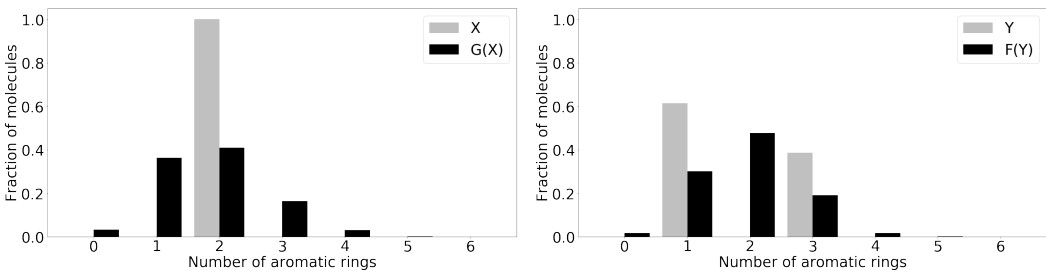

Figure 2: Distributions of the number of aromatic rings in $X$, $G(X)$, $Y$, and $F(Y)$. Identity mappings are not included in the figures.

In Table 1 we show the success rates for the tasks of performing structural transformations of molecules. The task of changing the number of aromatic rings is more difficult than changing the presence of halogen moieties. In the former the transition between $X$ (with 2 rings) and $Y$ (with 1 or 3 rings, cf. Fig. 2) is more than a simple addition/removal as it is in the other case. This is reflected in the success rates which are higher for the halogen moieties task. In the dataset used to construct the latent space (ZINC-250K) 64.9 % molecules do not contain any halogen moiety, whereas the remaining 35.1 % contain one or more halogen moieties. This imbalance might be the reason for the higher success rate in the task of removing halogen moieties ($Y \to F(Y)$).

To confirm that the generated molecules are close to the starting ones, we show in Fig. 3 distributions of their Tanimoto similarities (using Morgan fingerprints). For comparison we also include distributions of the Tanimoto similarities between the starting molecule and a random molecule from the ZINC-250K dataset. The high similarities between the generated and the starting molecules show that our procedure is neither a random sampling from the latent space, nor a memorization of the manifold in the latent space with the desired property value. We also visualize the molecules, which after transformation are the most similar to the starting molecules in Fig. 4.

## 4.2 CONSTRAINED MOLECULE OPTIMIZATION

As our main task we optimize the desired property under the constraint that the similarity between the original and the generated molecule is higher than some fixed threshold. This is a more realistic scenario in drug discovery, where the development of new drugs usually starts with known molecules such as existing drugs (Besnard et al., 2012). Here, we maximize the penalized logP coefficient and use the Tanimoto similarity with the Morgan fingerprint (Rogers & Hahn, 2010) to define the threshold of similarity, $sim(m, m') \geqslant \delta$. We compare our results with Jin et al. (2018) and You et al. (2018).

In our optimization procedure each molecule (given by the latent space coordinates $x$) is fed into the generator to obtain the 'optimized' molecule $G(x)$. The pair $(x, G(x))$ defines what we call 'optimization path' in the JT-VAE latent space. To be able to make a comparison with Jin et al. (2018) we start the procedure from the 800 molecules with the lowest values of penalized logP in

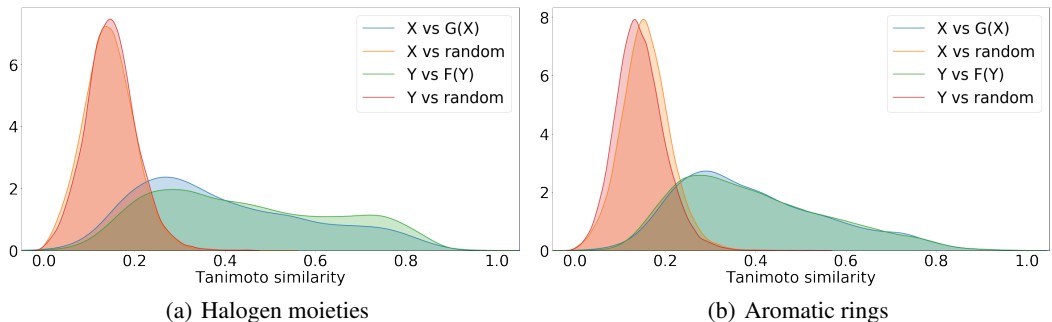

(a) Halogen moieties

(b) Aromatic rings

Figure 3: Density plots of Tanimoto similarities between molecules from $Y$ (and $X$) and their corresponding molecules from $F(Y)$ (and $G(X)$). Similarities between molecules from $Y$ (and $X$) and random molecules from ZINC-250K are included for comparison. Identity mappings are not included. The distributions of similarities related to transformations given by $G$ and $F$ show the same trend.

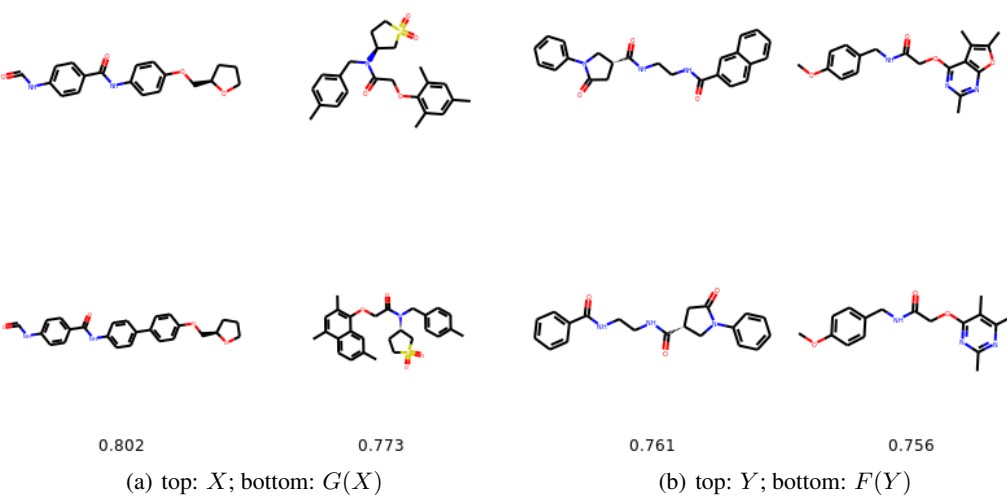

0.802          0.773          0.761          0.756

(a) top: $X$; bottom: $G(X)$          (b) top: $Y$; bottom: $F(Y)$

Figure 4: The most similar molecules with changed number of aromatic rings. In the top row we show the starting molecules, whereas in the bottom row we show the generated molecules. Below we provide the Tanimoto similarities between the molecules.

ZINC-250K. To allow for a fair comparison with Jin et al. (2018) (where $K = 80$ gradient ascent steps are made), we decode molecules from $K = 80$ points along the path from $x$ to $G(x)$ (in equal steps).

From the resulting set of $K$ molecules we report the molecule with the highest penalized logP score that satisfies the similarity constraint. A modification succeeds if one of the decoded molecules satisfies the constraint and is distinct from the starting one.

We show the results in Table 2. In the task of optimizing penalized logP of *drug-like* molecules, our method significantly outperforms the previous results in the mean improvement of the property. It achieves a comparable mean similarity in the constrained scenario (for $\delta > 0$). The success rates are comparable for $\delta = 0, 0.2$, whereas for the more stringent constraints ($\delta = 0.4, 0.6$) our model has lower success rates. Note that comparably high improvements of penalized logP can be obtained using reinforcement learning (You et al., 2018). However, the resulting optimized molecules are not druglike, e.g., they have a very low quantitative estimate of drug-likeness scores (Bickerton et al.,

Table 2: Results of constrained optimization for JT-VAE (Jin et al., 2018) and our Mol-CycleGAN.

| | JT-VAE | | | Mol-CycleGAN | | |
|---|---|---|---|---|---|---|
| Delta | Improvement | Similarity | Success | Improvement | Similarity | Success |
| 0 | $1.91 \pm 2.04$ | $\mathbf{0.28} \pm 0.15$ | 97.5% | $\mathbf{8.30} \pm 1.98$ | $0.16 \pm 0.09$ | $\mathbf{99.75\%}$ |
| 0.2 | $1.68 \pm 1.85$ | $\mathbf{0.33} \pm 0.13$ | $\mathbf{97.1\%}$ | $\mathbf{5.79} \pm 2.35$ | $0.30 \pm 0.11$ | 93.75% |
| 0.4 | $0.84 \pm 1.45$ | $0.51 \pm 0.10$ | $\mathbf{83.6\%}$ | $\mathbf{2.89} \pm 2.08$ | $\mathbf{0.52} \pm 0.10$ | 58.75% |
| 0.6 | $0.21 \pm 0.75$ | $\mathbf{0.69} \pm 0.06$ | $\mathbf{46.4\%}$ | $\mathbf{1.22} \pm 1.48$ | $\mathbf{0.69} \pm 0.07$ | 19.25% |

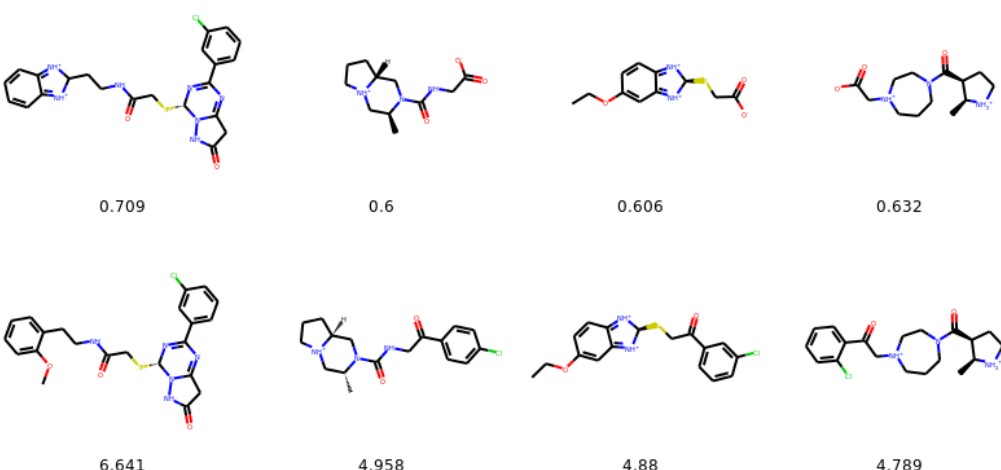

Figure 5: Molecules with the highest improvement of the penalized logP for $\delta \geqslant 0.6$. In the top row we show the starting molecules, whereas in the bottom row we show the generated molecules. Upper row numbers indicate Tanimoto similarities between the starting and the generated molecule. Improvement in the score is given in at the bottom.

2012) even in the constrained optimization scenario. In our method (as well as in JT-VAE) drug-likeness is achieved 'by construction' and is a feature of the latent space obtained by training the variational autoencoder on molecules from ZINC (which are druglike).

### 4.3 Unconstrained molecule optimization

Our architecture is tailor made for the scenario of constrained molecule optimization. However, as an additional task, we check what happens when we iteratively use the generator on the molecules being optimized, which leads to diminishing of similarity between the starting molecules and those in consecutive iterations. For the present task the set $X$ needs to be a sample from the entire ZINC-250K, whereas the set $Y$ is chosen as a sample from the top-20% of molecules with the highest value of penalized logP. Each molecule is fed into the generator and the corresponding 'optimized' molecule is obtained. The generated molecule is then treated as the new input for the generator. The process is repeated $K$ times and the resulting set of molecules is $\{G(x), G(G(x)), ... \}$. Here, as in the previous task and as in Jin et al. (2018) we start the procedure from the 800 molecules with the lowest values of penalized logP in ZINC-250K.

The results of our unconstrained molecule optimization are shown in Figure 6. In Fig. 6(a) and (c) we observe that consecutive iterations keep shifting the distribution of the objective (penalized logP) towards higher values. However, the improvement from further iterations is decreasing. Interestingly, the maximum of the distribution keeps increasing (although in somewhat random fashion). After 10-20 iterations it reaches the high values observed from molecules which are not druglike in You et al. (2018) (obtained with RL). In our case the molecules with the highest penalized logP after many iterations also become non-druglike – see Appendix D for a list of compounds with the max-

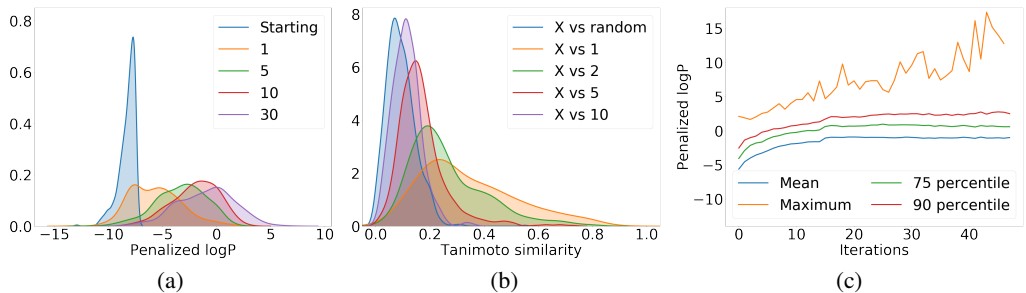

Figure 6: Results of iterative procedure of the unconstrained optimization. (a) Distribution of penalized logP in the starting set and after $K = 1, 5, 10, 30$ iterations. (b) Distribution of the Tanimoto similarity between the starting molecules $X$ and random molecules from ZINC-250K, as well as those generated after $K = 1, 2, 5, 10$ iterations. (c) Plot of the mean value, quantiles (75th and 90th), and the maximum value of penalized logP as a function of the number of optimization iterations.

imum values of penalized logP in our iterative optimization procedure. This lack of drug-likeness is related to the fact that after performing many iterations, the distribution of coordinates of our set of molecules in the latent space goes far away from the prior distribution (multivariate normal) used when training the JT-VAE on ZINC-250K. In Fig. 6(b) we show the evolution of the distribution of Tanimoto similarities between the starting molecules and those obtained after $K = 1, 2, 5, 10$ iterations. We also show the similarity between the starting molecules and random molecules from ZINC-250K. We observe that after 10 iterations the similarity between the starting molecules and the optimized ones is comparable to the similarity to random molecules from ZINC-250K. After around 20 iterations the optimized molecules become less similar to the starting ones than random molecules from ZINC-250K.

## 5 CONCLUSIONS

In this work, we introduced Mol-CycleGAN – a new model based on CycleGAN that can be used for the *de novo* generation of molecules. The advantage of the proposed model is the ability to learn transformation rules from the *sets* of compounds with desired and undesired values of the considered property. The model operates in the latent space trained by another model – in our work we use the latent space of JT-VAE. The model can generate molecules with desired properties – both structural and physicochemical. The generated molecules are close to the starting ones and the degree of similarity can be controlled via a hyperparameter. In the task of constrained optimization of drug-like molecules our model significantly outperforms previous results. In future work we will extend the approach to multi-parameter optimization of molecules using StarGAN (Choi et al., 2017). It would also be interesting to test the model on cases where a small structural change leads to a drastic change in the property (e.g. on the so-called activity cliffs), which are hard for other approaches. Another interesting direction is the application of the model to working on text embeddings, where the $X$ and $Y$ sets could be characterized, e.g., by different sentiment.

ACKNOWLEDGEMENTS

We would like to thank [name redacted] for their helpful comments and for fruitful discussions.

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

## A ARCHITECTURE OF MODELS

All networks are trained using the Adam optimizer (Kingma & Ba, 2014) with learning rate 0.0001. During training we use batch normalization (Ioffe & Szegedy, 2015). As the activation function we use leaky-ReLU with $\alpha = 0.1$. In experiments from sections 4.1, 4.2 the models are trained for 100 epochs and in experiments from 4.3 for 300 epochs.

### A.1 FOR EXPERIMENTS IN SECTIONS 4.1, 4.2

- **Generators** are built of one fully connected residual layer, followed by one dense layer. All layers contain 56 units.

- **Discriminators** are build of 6 dense layers of the following sizes: 56, 42, 28, 14, 7, 1 units.

### A.2 FOR EXPERIMENTS IN SECTION 4.3

- **Generators** are built of four fully connected residual layers. All layers contain 56 units.

- **Discriminators** are build of 7 dense layers of the following sizes: 48, 36, 28, 18, 12, 7, 1 units.

## B COMPOSITION OF DATASETS

### B.1 DATASET SIZES

In table 3 we show the sizes of the datasets used for training (i.e. the number of molecules in each of them). In all experiments we use a separate dataset for training the model (train $X$ and train $Y$) and a separate, non-overlapping one for evaluating the model (test $X$ and test $Y$). In Experiments 4.2 and 4.3 the model was used to perform optimization and only the generator $G$ was used, hence no test $Y$ set was required.

Table 3: Dataset sizes for experiments in section 4

| Dataset | Halogen moieties | Aromatic rings | Experiment 4.2 | Experiment 4.3 |
|---|---|---|---|---|
| train $X$ | 75000 | 80000 | 80000 | 80000 |
| test $X$ | 86899 | 18220 | 800 | 800 |
| train $Y$ | 75000 | 80000 | 80000 | 24946 |
| test $Y$ | 12556 | 43193 | - | - |

### B.2 DISTRIBUTION OF THE SELECTED PROPERTY

In the experiment on halogen moieties, the set $X$ always (i.e., both in train- and test-time) contains molecules without halogen moieties, and the set $Y$ always contains molecules with halogen moieties. In the dataset used to construct the latent space (ZINC-250K) 64.9 % molecules do not contain any halogen moiety, whereas the remaining 35.1 % contain one or more halogen moieties.

In the experiment on aromatic rings, the set $X$ always (i.e., both in train- and test-time) contains molecules with 2 rings, and the set $Y$ always contains molecules with 1 or 3 rings. The distribution of the number of aromatic rings in the dataset used to construct the latent space (ZINC-250K) is shown in Fig. 7 along with the distribution for $X$ and $Y$.

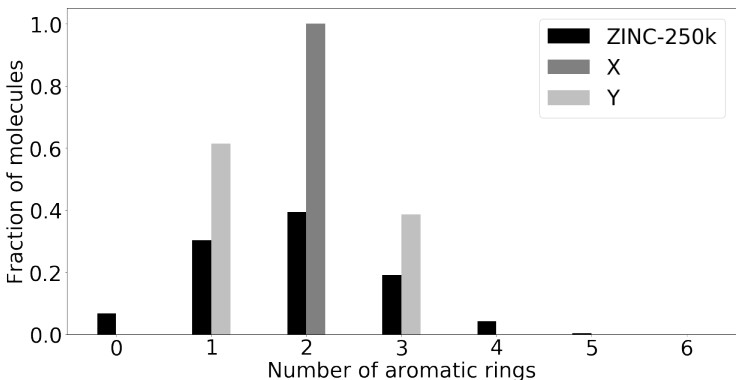

Figure 7: Number of aromatic rings in ZINC-250K and in the sets used in the experiment on aromatic rings (Section 4.1)

For the molecule optimization tasks we plot the distribution of the property being optimized (penalized logP) in Figs. 8 (constrained optimization) and 9 (unconstrained optimization).

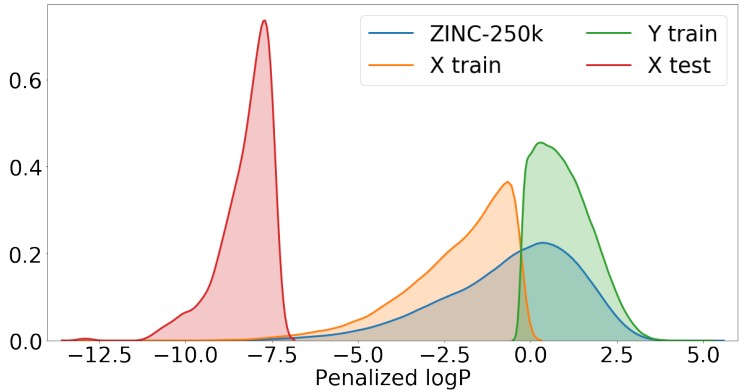

Figure 8: Distribution of penalized logP in ZINC-250K and in the sets used in the task of constrained molecule optimization (Section 4.2). Note that the sets $X$ train and $Y$ train are non-overlapping (they are a random sample from ZINC-250K split by the median). $X$ test is the set of 800 molecules from ZINC-250K with the lowest values of penalized logP.

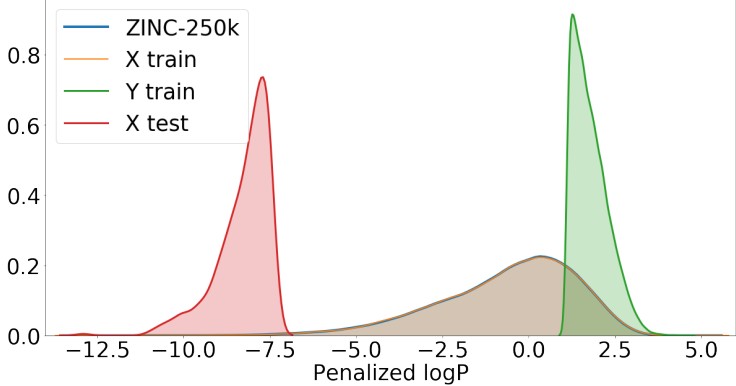

Figure 9: Distribution of penalized logP in ZINC-250K and in the sets used in the task of unconstrained molecule optimization (Section 4.3). Note that the set $X$ train is a random sample from ZINC-250K, and hence the same distribution is observed for the two sets.

# C   MOLECULAR PATHS FROM OPTIMIZATION EXPERIMENTS IN SECTIONS 4.2 AND 4.3

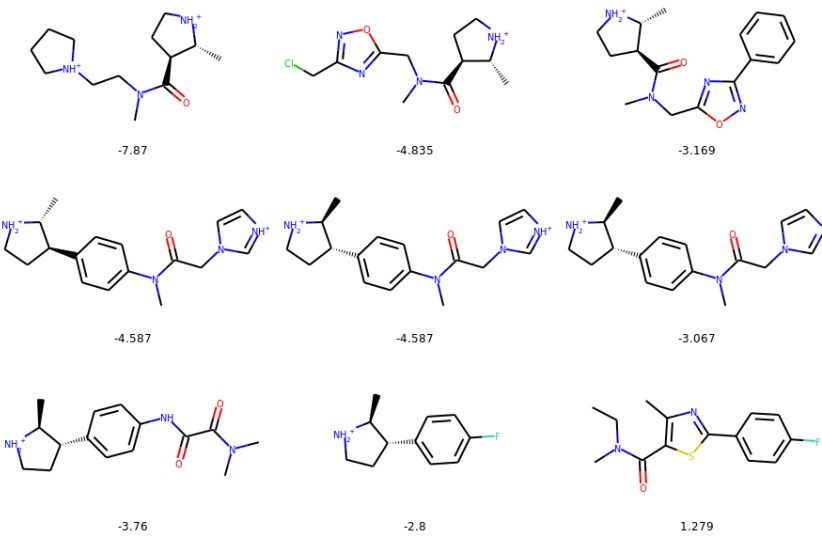

Figure 10: Evolution of a selected exemplary molecule during constrained optimization. We only include the steps along the path where a change in the molecule is introduced. We show penalized logP below the molecules.

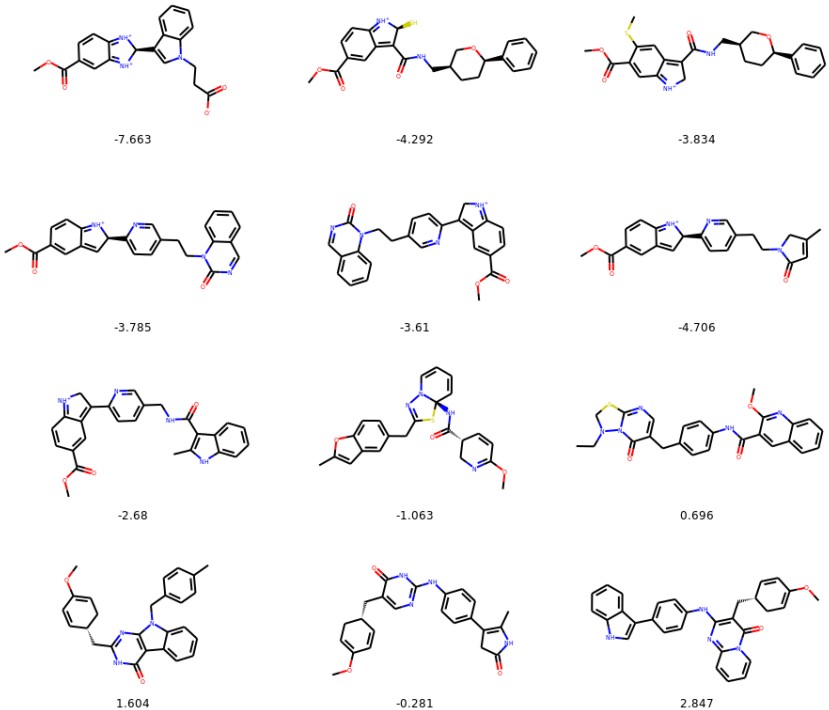

Figure 11: Evolution of a selected exemplary molecule during constrained optimization. We only include the steps along the path where a change in the molecule is introduced. We show penalized logP below the molecules.

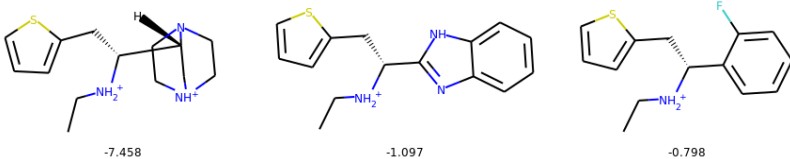

Figure 12: Evolution of a selected exemplary molecule during constrained optimization. We only include the steps along the path where a change in the molecule is introduced. We show penalized logP below the molecules.

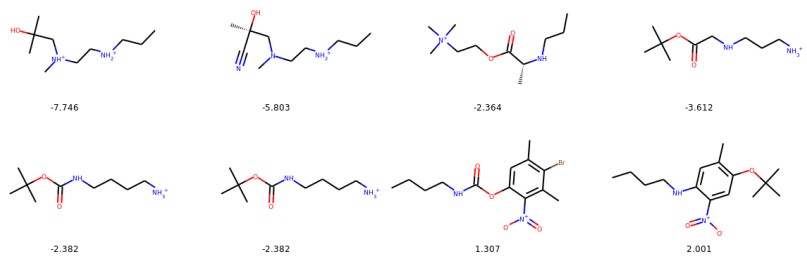

Figure 13: Evolution of a selected molecule during consecutive iterations of unconstrained optimization. We show penalized logP below the molecules.

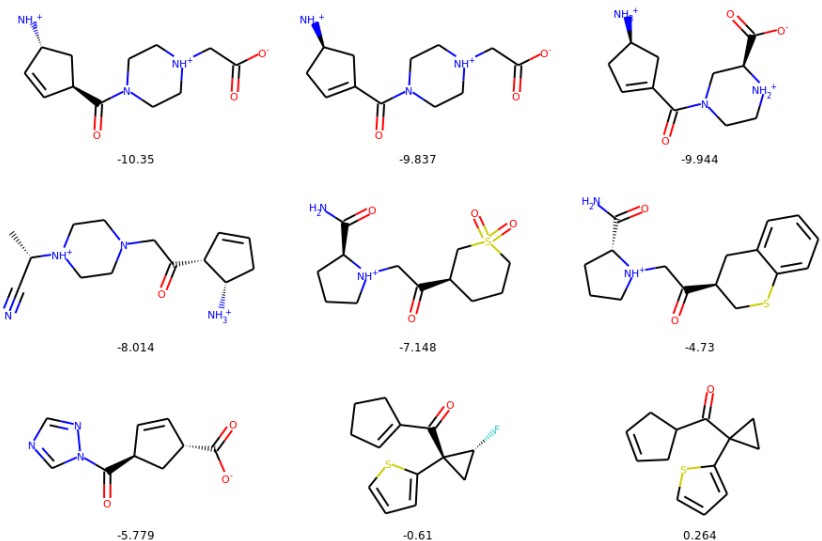

Figure 14: Evolution of a selected molecule during consecutive iterations of unconstrained optimization. We show penalized logP below the molecules.

## D MOLECULES WITH THE HIGHEST PENALIZED LOGP

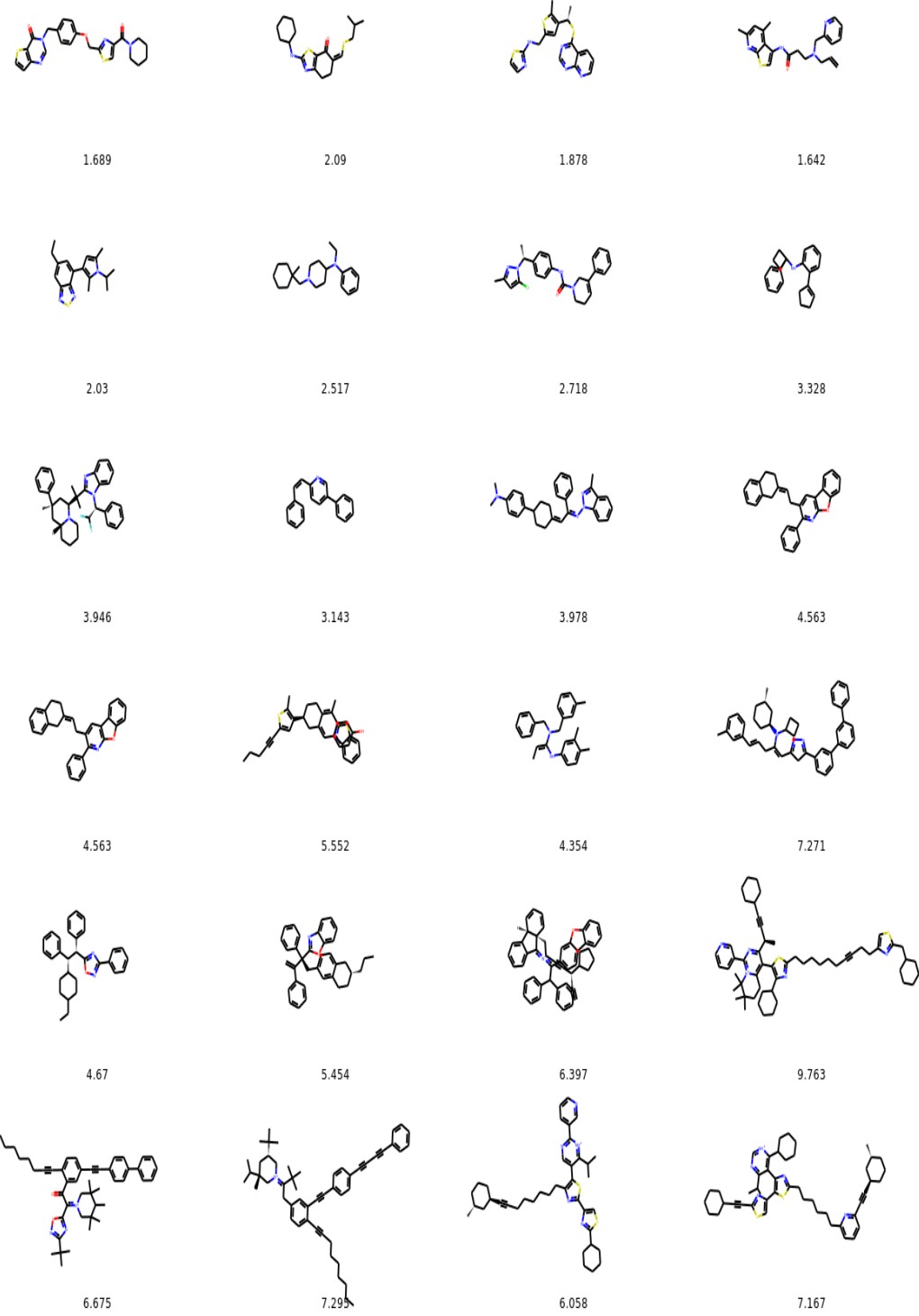

Figure 15: Molecules with the highest penalized logP in the set being optimized for iterations 1-24 for unconstrained molecule optimization. We show penalized logP below the molecules.

