# OpenReview forum: "Mol-CycleGAN - a generative model for molecular optimization"
_ICLR.cc/2019/Conference_

### Official Review · AnonReviewer1 · 2018-11-02
**Not good enough**

**Rating:** 4
**Confidence:** 3

**Review:**

A strand of papers has been proposed for optimizing molecules for a given property using different forms of variational autoencoders as a tool to extract embedding for molecules. This paper is also in this direction by applying Cycle-GAN [Zhu et al., 2017] in the latent space learnt from junction-tree VAE [Jin et.al, 2018], aiming to learn a translation function (in the latent space), from the set of molecules without the interested property to the set of molecules with the property.

The paper is well written and the contribution is more on the application side, although combining the above mentioned two gradients are indeed novel. That being said, I wish the authors can bring more perspectives from chemists or biologists which can justify and support the application setting.

Since the focus is molecular optimization, I paid special attention to the results in Section 4.2. While the improvement regarding to the interested property (logP) is a pleasure to see, the drop of the success rate is also significant comparing to [Jin et.al, 2018], which undermines the significance of the results, especially given the increased complexity. Another issue is the requirement of a large set of molecules with desired property. This further restricts the applicability of the method. How to solve the cold start issue will be critical in this setting and this is not mentioned in the paper. Thirdly, combining two existing components is ok but not enough novelty from my point of view. Considering the high acceptance bar of ICLR, I will not accept this paper.

Detailed comments:
1. In Section 4.2, how similar the generated molecules are to the ones already in the Y_train?
2. In Section 4.2, G(X) map to Y, what does it mean to apply G(G(X))? Do you decode G(X) to a molecule first and then feed into the encoder to apply G (as in Section 4.3)? If not, G is not supposed to learn the transition from Y to X. Will you always get exactly the same X when apply F(G(X))? Can the sequence be G(X), G(F(G(X))), …?
3. In Section 4.3, is it always that later step gives better logP? It seems so from Figure 6 for 1-30 iterations, how about later 30-80 iterations? If so, for the generated molecules, the method seems have a tension between the similarity to the original molecules and the level of the desired property. Can you comment on how important, in practice, the similarity matters?
4. Following 3, \lambda_2 seems directly affect the balance of the tension and it should be studied in more detail.
5. How about reproducibility, will the data and code published?

---

### Official Review · AnonReviewer3 · 2018-11-02
**An application of CycleGAN to molecule transformations**

**Rating:** 4
**Confidence:** 4

**Review:**

Main idea:
This paper proposes to use CycleGAN to generate a molecule with a desired property given an initial molecule without that property.  This is a task appearing in drug design. Specifically, the training set consists of two sets of molecules with and without a desired property. For each molecule, a representation/embedding is obtained, which is the latent code of the JT-VAE. Two generators/transformers are introduced to perform transformation from the representation of one domain to that of another domain. The training involves a weighted sum of several losses: the LSGAN loss, the cyclic consistency loss and the reconstruction loss, which are exactly the same losses used in the CycleGAN paper.  Experiments are conducted on ZINC-250K dataset and evaluated in terms of structural modifications and molecule optimization.

Comments:
This is a good application paper using machine learning techniques. However, I don't think ICLR is a suitable venue for this paper since the improvements and contributions are more on molecule optimization side. I suggest that the authors extend the technical part of CycleGAN with more background introduction and submit the paper to a bioinformatics journal.

---

### Official Review · AnonReviewer2 · 2018-11-07
**Promising results, unfortunately not strong enough for main track**

**Rating:** 4
**Confidence:** 5

**Review:**

The paper presents an approach for optimizing molecular properties, based on the application of CycleGANs to variational autoencoders for molecules. A recently proposed domain-specific VAE called Junction Tree VAE (JT-VAE) is employed. The optimization of molecules is an important problem for example in drug discovery or materials design, and the development of machine learning approaches for it is therefore an important application.


On the positive side, this reviewer would like to highlight the structural transformation task, which is an interesting addition. The presentation is mostly clear, and there is an improvement in benchmark scores compared to the baseline JT-VAE.

On the negative side, this reviewer would argue the paper is a bit incremental, as it appears to be “just” a combination of two models without a clear motivation from a molecular design perspective – why is the model e.g. better (motivated) than the model by You et al, the MOLGAN by DeCao and Kipf, or the CGVAE by Liu et al?
 Also, the number of benchmark experiments is small compared to more rigorous evaluation studies, e.g. by Olivecrona et al. https://doi.org/10.1186/s13321-017-0235-x

Furthermore, a lot of (labeled) training examples for the respective properties seem to be needed, whereas in practical drug discovery, often only very small datasets are available.



Questions:

How exactly is the similarity constraint enforced?

Why do the authors use the JT-VAE and not the SD-VAE or CGVAE?

Can the authors explain why they think penalized logP is relevant to drug discovery? While the optimization of physicochemical properties alone is certainly important in a multi-objective setting, this reviewer is not entirely sure that physicochemical property optimization alone is a problem that often occurs in drug discovery.

Why are the values for the RL model by You et al. 2018 not shown in Table 2, even though the authors state that they have done the comparison? If those molecules are not drug-like, it would be good to see the comparison, and show the respective molecules. Keep in mind that you are not asking the models to produce drug-like molecules here! The RL model therefore seems to generate exactly what it is asked (rewarded) for. Is it a fair comparison that you expect it to do something else?

Does the requirement to define the sets X and Y restrict the capacity of the model to binary class membership, as in “good” and “bad”? Does this imply the model cannot extrapolate into e.g. continuous parameter ranges unseen during training?

Provocative statement this reviewer would be keen to hear the authors’ (and co-reviewers) opinion on: It seems that there are considerable efforts needed in getting VAEs for discrete data such as graphs to work, namely the drastic restriction of the molecular graph space via grammars, CGVAE, or the JT setting (which builds on a heuristically defined vocabulary), whereas RL models even get decent results with simple SMILES-RNNs (see e.g. the work by Merk et al, who have even successfully made and tested their results in the lab!  https://doi.org/10.1038/s42004-018-0068-1 and https://doi.org/10.1002/minf.201700153 ) Do the author have an intuition into why this might be the case?


Regarding the proposed application of GANs to text generation, this reviewer would suggest to have a look at this recent (as in yesterday) paper https://arxiv.org/pdf/1811.02549.pdf



Overall, this reviewer believes the paper would be a good ICLR workshop paper or, after a bit more technical details on the models, a decent chemoinformatics paper, but unfortunately this reviewer is not convinced that in the current form the results are significant or general enough in terms of ML to warrant acceptance in the main ICLR track.

---

### Meta-Review · Area_Chair1 · 2018-12-09
**Fundamentally sensible idea, but too incremental and application-focused**

**Confidence:** 4
**Recommendation:** Reject

**Metareview:**

This paper introduces a variant of the CycleGAN designed to optimize molecular graphs to achieve a desired quality.  The work is reasonably clear and sensible, however it's of limited technical novelty, since it's mainly just combining two existing techniques.  Overall its specificity and incrementalness make it not meet the bar.